# Trunk Range of Motion Is Related to Axial Rigidity, Functional Mobility and Quality of Life in Parkinson’s Disease: An Exploratory Study

**DOI:** 10.3390/s20092482

**Published:** 2020-04-27

**Authors:** Roberto Cano-de-la-Cuerda, Lydia Vela-Desojo, Marcos Moreno-Verdú, María del Rosario Ferreira-Sánchez, Yolanda Macías-Macías, Juan Carlos Miangolarra-Page

**Affiliations:** 1Department of Physiotherapy, Occupational Therapy, Rehabilitation and Physical Medicine, Health Sciences Faculty, Rey Juan Carlos University, Alcorcón, 28922 Madrid, Spain; roberto.cano@urjc.es (R.C.-d.-l.-C.); juan.miangolarra@urjc.es (J.C.M.-P.); 2Neurology Service. Division of Movements Disorders, Hospital Universitario Fundación Alcorcón (HUFA), Alcorcón, 28922 Madrid, Spain; LVela@fhalcorcon.es (L.V.-D.); ymacias@cop.es (Y.M.-M.); 3Asociación Parkinson Madrid, 28014 Madrid, Spain; 4Department of Radiology, Rehabilitation and Physiotherapy, Complutense University of Madrid, 28040 Madrid, Spain; masanc23@ucm.es; 5Rehabilitation Unit, Hospital Universitario de Fuenlabrada, Fuenlabrada, 28942 Madrid, Spain

**Keywords:** axial rigidity, health related quality of life, Parkinson’s disease, range of motion

## Abstract

**Background:** People with Parkinson’s disease (PD) present deficits of the active range of motion (ROM), prominently in their trunk. However, if these deficits are associated with axial rigidity, the functional mobility or health related quality of life (HRQoL), remains unknown. The aim of this paper is to study the relationship between axial ROM and axial rigidity, the functional mobility and HRQoL in patients with mild to moderate PD. **Methods:** An exploratory study was conducted. Non-probabilistic sampling of consecutive cases was used. Active trunk ROM was assessed by a universal goniometer. A Biodex System isokinetic dynamometer was used to measure the rigidity of the trunk. Functional mobility was determined by the Get Up and Go (GUG) test, and HRQoL was assessed with the PDQ-39 and EuroQol-5D questionnaires. **Results:** Thirty-six mild to moderate patients with PD were evaluated. Significant correlations were observed between trunk extensors rigidity and trunk flexion and extension ROM. Significant correlations were observed between trunk flexion, extension and rotation ROM and GUG. Moreover, significant correlations were observed between trunk ROM for flexion, extension and rotations (both sides) and PDQ-39 total score. However, these correlations were considered poor. **Conclusions:** Trunk ROM for flexion and extension movements, measured by a universal goniometer, were correlated with axial extensors rigidity, evaluated by a technological device at 30°/s and 45°/s, and functional mobility. Moreover, trunk ROM for trunk flexion, extension and rotations were correlated with HRQoL in patients with mild to moderate PD.

## 1. Introduction

The decreased amplitude of joint movement is one of the characteristic symptoms in Parkinson’s disease (PD) [1,2]. This does not only affect distal joints but is also present in the trunk [3,4,5], producing an impact on the patients’ performance in their daily activities and function [6,7]. Trunk mobility is a key component of physical therapy treatment in PD patients, so an accurate measurement of the range of motion (ROM) of the trunk is often considered an essential component in a rehabilitation context [8]. Limitations of the ROM within the spine can create compensatory mechanisms which result in a greater motion requirement in other body parts and lead to muscular asymmetries and increased injury risk [9].

One of the possible explanations regarding these ROM deficits is the presence of an intrinsic muscle stiffness at the local level [6,10,11]. Rigidity is one of the most frequent symptoms in PD [12,13]. Clinically, it is defined as the increased resistance to the passive stretch of a given muscle or muscle group [14]. In fact, this phenomenon is present both distally and axially [15] and both at rest and with voluntary movement [16]. There have been found associations between trunk rigidity and the functional mobility [17] and with Health-Related Quality of Life (HRQoL) [18]. Therefore, axial muscle stiffness could be the main cause of the trunk ROM deficits present in PD. Patients could present lesser articular amplitude due to a local musculature, which has an excessively high tone, thus limiting the complete articular motion.

Although these hypotheses seem plausible, limited research has been developed to explore the relationships between these clinical phenomena. Given the importance of trunk mobility and the relationship between axial rigidity with risk of falls [17,19] and activities of daily living [20], studies are needed to identify limitations of the ROM in PD patients.

The objective measurement of rigidity is difficult and complicated in the clinical context. Instrumentalized tools with high costs and complex procedures are needed in order to obtain reliable and quantitative data (such as servomotors, portable devices, and others), since clinical scales and methods are subjective in their nature and present biases [21]. For example, technological methods have been developed to measure muscle tone in PD [22]. In previous studies, several authors have used technological evaluations with isokinetic dynamometers to examine trunk rigidity in PD subjects, thanks to the mobilization of the trunk at desired speeds throughout the available ROM, recording the information relative to the offered resistance as an objective measure of rigidity [15,18,23,24]. Other technologies like inertial sensors and biomechanical and neurophysiological muscle measurements have been described [25], but all these evaluations techniques are rarely available in daily clinical practice since equipment is expensive, and its use is time consuming and requires complex data analysis.

Nevertheless, in clinical practice, conventional goniometry assessment is used by therapists to evaluate the active and passive ROM of the trunk due to its advantages, such as ease of use, cost and intra and inter-rater reliability [26]. If the ROM of the trunk and axial rigidity are significantly associated, goniometric measurements could give an idea to the clinician of the patient’s axial muscle tone, in an easier and faster way, which could be relevant to implement therapeutic actions.

On the other hand, axial rigidity seems to be correlated with the functional mobility and HRQoL, as our research team have suggested in previous works [17,18]. Therefore, to explore the relationship between the trunk ROM and these two constructs seems suitable, in the context of a possible interrelationship of these relevant variables for the patient with PD.

Thereby, our principal aim is to analyse the existing relationships between the ROM of the trunk, in all its movements, and axial rigidity, in patients with PD. Secondarily, the relationships between the ROM of the trunk and the functional mobility and HRQoL were explored.

## 2. Materials and Methods

### 2.1. Study Design

A cross-sectional study was conducted at a Motion Analysis Laboratory located at a University Department. Non-probabilistic sampling of consecutive cases was used. The study was approved by a local Ethical Committee, and informed consent was obtained from each subject in accordance with the 1964 Declaration of Helsinki. All patients gave their informed consent prior to their inclusion in the study. All assessments were performed by the same investigator expert in movement disorders. The same environmental conditions were maintained during all assessments to limit the influence of external factors.

### 2.2. Participants

All patients were recruited from a Neurology Service (Hospital Universitario Fundación Alcorcón, Madrid, Spain), with a diagnosis of idiopathic PD [27].

The inclusion criteria were as follows: (1) stages I-III of the Hoehn and Yahr scale, (2) no diseases that affect the trunk and lower limbs, (3) stable on anti-Parkinsonian medication. The study exclusion criteria were as follows: (a) the diagnosis of diseases other than PD; (b) patients with dementia, established by a Mini-Mental State Examination (MMSE) [28] score ≤ 23 points; (c) static postural abnormalities (such as camptocormia, pisa syndrome, antecollis, or others); (d) Psychiatric problems; (e) patients with depression established by a Beck’s Depression Inventory score ≤ 13 points. Patients with psychiatric or depression diagnosis could not follow all instructions during the conventional and technological assessments or did not fill the questionnaires correctly.

### 2.3. Procedure

All tests were performed by the same rater within 1–3 h of the administration of anti-Parkinsonian medication. All patients were assessed by the Hoehn and Yahr scale and the Unified Parkinson’s Disease Rating Scale-Motor Scale (UPDRS-III) [29] to evaluate stage severity and motor impairment, respectively. Functional status, related to the ability to perform activities of daily living (ADLs), in terms of speed and independence, was measured with the Schwab and England scale [30].

#### 2.3.1. Trunk ROM by Conventional Assessment

A universal goniometer was used. The measurement of the active ROM of the trunk was performed according to the method described by Norkin and White [26] and Chertman et al. [31] for flexion and extension movements (Figure 1).

For trunk rotation, PD patients were instructed to place in sitting position with their feet on the floor, and knees and hips in 90° of flexion. The goniometer was positioned with the axis fixed in the transverse plane at the level of T1-T2, following the measure recommendations [32] (Figure 2).

For trunk lateral flexion, PD patients began the test in an upright standing position, with the knees completely extended. Then, upon a verbal command from the examiner, they did slow and gradual lateral flexion movements in a maximum amplitude, where the goniometric measurement was made. To evaluate lateral flexion, the first sacral vertebra was taken as the axis fixed reference point. The stationary arm of the universal goniometer was pointed perpendicular to the floor, while the mobile arm remained pointing to the spinous process of the seventh cervical vertebra [26] (Figure 3).

The active ROM was measured three times for trunk flexion, extension, lateral flexion (both sides) and rotation (both sides) movements, by the same examiner. An average of the measurements of each movement was used for the analysis. At the end of the measurement, the participant was instructed to return to the starting position where the universal goniometer was removed and repositioned before the following measurement. Participants were allowed to rest while the examiner entered the measurement into a spreadsheet.

#### 2.3.2. Trunk Rigidity by Technological Assessment

Trunk extension-flexion component of a Biodex System isokinetic dynamometer was used to measure the rigidity of the trunk. Subjects were instructed to rest their body comfortably against the back of the trunk dynamometer component. The investigator then measured the anatomical zero, which was then used as a reference point to measure trunk muscle tone over a range of 50°, from 30° trunk extension to 80° flexion (S: 30-50-80) [8,9,10].

All subjects were examined prior to the test to ensure that they had the appropriate passive range of trunk movement. The continuous passive mode of Biodex System with three angular speeds (30, 45 and 60°/s) was used to assess trunk muscle tone. To ensure subject safety, the Biodex dynamometer was calibrated using the “speed calibration” programme. If subjects felt any discomfort during the test, the examiner could stop the machine immediately by pressing a safety switch. During the test, the subject’s trunk was passively moved into flexion then extension, and muscle resistance was recorded over the range. For each speed in every movement, one practice trial was given, and five trials were recorded with the patient being instructed to relax. Each trial was followed by 30 s of rest. To minimise possible errors due to the inertia of the subject’s weight, the resistance measurements of the first trial were discarded, and the measurements from the remaining four trials were analysed. This protocol was validated in previous studies by our research team [8,9,10].

The overall trunk extensors rigidity and trunk flexors rigidity were expressed as work done (Nm°/Body Weight) (W/BW). The resistance to passive trunk flexion represented trunk extensor muscle tone, while passive trunk extension represented trunk flexor muscle tone.

#### 2.3.3. Functional Mobility Assessment

The Get Up and Go (GUG) test is a measure of functional mobility in PD patients. For the assessments, the patient is asked to stand up from a standard chair and walk a distance of 3 m, turn around and walk back to the chair and sit down again. Mobility is rated on a 5-point scale, ranging from “normal” to “severely abnormal” according to the observation of a clinician [33].

#### 2.3.4. Health-Related Quality of Life (HRQoL) Assessment

HRQoL was assessed with the 39-item Parkinson’s Disease Questionnaire (PDQ-39) and with the Euro-Quality of life-5 dimensions (EuroQol-5D). PDQ-39 is a specific instrument to assess HRQoL in PD and comprises 39 questions related to its frequency in the previous month. Scores ranging from 0 to 100 and higher scores indicate worst HRQoL [34].

The EuroQol-5D is a generic HRQoL questionnaire. It is a self-administered instrument and includes a visual analogue scale on which patients rate their own health between 0 and 100. Higher scores indicate better HRQoL [35].

### 2.4. Data Analysis

The Statistical Package for the Social Sciences (SPSS) program was used. Spearman correlation coefficient (r) was used to establish the relationship between the ROM of the trunk, axial muscle stiffness, functional mobility and HRQoL. Correlation coefficients of 0.00 to 0.49 were interpreted as poor, from 0.50 to 0.79 as moderate and ≥0.8 as excellent [36]. p value <0.05 was considered as a significance level.

## 3. Results

Thirty-six PD patients participated in the study (29 males). The mean age of the sample was 62 ± 11 years. The left side was the side most affected in 40% of the sample, while the right side was the most affected in the 60% of the sample. All of them were classified between I–III in the Hoehn and Yahr scale and reached 80%–100% scores in the Schwab and England Scale. Mean values of disease duration and disease severity are described in Table 1.

Mean values of the ROM of the trunk, the rigidity of the trunk, functional mobility and HRQoL are described in Table 2.

Correlations between the ROM of the trunk, axial rigidity and GUG are described in Table 3. Significant correlations were only observed between the rigidity of the extensors of the trunk at 30°/s and 45°/s and the trunk flexion and extension ROM (*r* = −0.687, *p* = 0.005; *r* = −0.534, *p* = 0.042; *r* = −0.510, *p* = 0.006; *r* = −0.410, *p* = 0.005, respectively). No other significant correlations were observed.

Significant correlations were observed between the ROM of the trunk for flexion, extension and rotation (both sides) and GUG scores (Table 3).

Statistically significant correlations were found between ROM and PDQ-39 total score only for trunk flexion, extension and rotations for both sides (Table 4). These correlations were considered poor. Moreover, a moderate correlation was observed between trunk ROM for extension and PDQ-39 total score. No correlations were observed between the ROM of the trunk at any movement and health status assessed by EuroQol-5D (Table 4).

## 4. Discussion

Goniometry is one of the most extended methods for the assessment of the ROM by physical therapists, due to its ease of use, cost effectiveness and reliability [26]. To our best knowledge, this is the first exploratory study investigating the relationship between the ROM of the trunk assessed by a goniometer and other relevant constructs in PD as axial muscle tone, assessed by an objective technological device, functional mobility and HRQoL. Our results showed significant correlations between flexor and extensor trunk movements assessed by a conventional measurement method of the ROM and trunk extensors rigidity in patients with mild to moderate PD. Moreover, our findings emphasise the importance of addressing spine ROM in patients with a clinical diagnosis of PD at early stages.

### 4.1. Range of Motion of the Trunk and Axial Rigidity

People with early PD exhibit less axial ROM compared with healthy controls [6,37,38]. This is observed by using instrumentalized methods such as dynamometers or motion capture analysis tools. This finding does not appear exclusively during the clinical evaluation of the ROM [6] but also, and what is more relevant, during functional activities like reaching [37], frontal gait or turning [7,39,40], and sit-to-stand [41]. Some of the factors that contribute to these impairments could be a decreased muscle torque production ability, disrupted motor planning, peripheral neuromuscular changes, altered characteristics of the noncontractile muscle elements, abnormal discharge characteristics of motor units, disuse weakness and muscular rigidity [6,41]. In this sense, to analyse how the trunk ROM deficits are associated with axial rigidity or bradykinesia seems pertinent.

In our study, we have found significant correlations between extensors rigidity at specific movement velocities (30°/s and 45°/s), and flexion and extension ROM deficits. These findings could be due to the fact that mostly of the ADL are performed in a specific trunk ROM, not total, and under speed movements considered until 45°/s. However, this contrasts with the evidence that people with PD exhibit axial ROM deficits in rotation, as measured by a triaxial dynamometer [6]. In our paper, we did not observe significant relationships between the flexor muscles hypertonia and trunk ROM. These findings could be related with the small sample size and exclusion criteria of patients with stages IV or V in the Hoehn and Yahr scale. Further, the testing sequence was the same for all the subjects, so a possible bias due to the test order (flexion, extension, rotation to right and left, lateral flexion to right and left) could be present. Moreover, a short range of motion to trunk flexion was tested (from 50° to 80°; only 30° to flexion movement). Therefore, a randomized assessment with the isokinetic dynamometer under low to medium speed movements and a wider range of motion should be implemented in future studies.

In short, our results showed only a partial relationship between trunk ROM and axial rigidity. These findings could be used in future studies to determine improvements in axial rigidity in a post-treatment period, using of a simple and feasible ROM assessment, as suggested in previously published studies about the effects of deep brain stimulation in trunk movements in PD patients [42,43].

### 4.2. Trunk Range of Motion and Functional Mobility

To determine risk of falls, three-dimensional computerized analysis may be considered the gold standard because it provides objective data about kinematic and kinetic parameters. Nevertheless, customized instrumentation is needed; the equipment is expensive; it takes longer time and requires complex data analysis, and it is not always available in clinical practice [44,45]; therefore, the most extended methods in the therapeutic environment are clinical tests. In this context, observational assessment scales, as the GUG scale, have been designed to indirectly measure this risk, since they evaluate deviations from normal patterns during functional mobility tasks such as sit-to-stand, walking and turning. Although this test does not directly measure the risk of falls due to its qualitative and subjective nature, it is recommended as a routine screening test following guidelines published by the American Geriatric Society and the British Geriatric Society [46] and the National Institute of Clinical Evidence (NICE) for gait and balance assessment in the prevention of falls [47].

In our study, the ROM of the trunk measured by goniometry was partially related to functional mobility assessed by the GUG in a sample of mild to moderate PD patients. Our results are consistent with other studies that have found ROM deficits in PD fallers in comparison with PD non-fallers and healthy controls, in other body parts such as the hip joint. Besides, these studies have used instrumentalized methods with three-dimensional motion analysis techniques [48].

Our findings suggest that an association exists between functional mobility and ROM of the trunk for the flexion, extension and rotation (both sides) movements. Furthermore, this association is stronger for the rotation movements (both sides). These associations make sense since these movements are used by the patient to perform the GUG. Thus, we did not find associations between the GUG and the lateral flexion ROM, probably because this test does not demand to the patient to make these movements and ROM for lateral flexions were not highly affected at this stage of the disease in our sample. However, to recuperate and/or maintain this specific articular joint amplitude should be an essential objective in the physical therapy programs for PD. Other instrumentalized methods could find associations between the functional mobility and the lateral flexion ROM, so more research is needed in this field.

### 4.3. Axial Impairments and Health Related Quality of Life

Axial impairments related to gait disorders, postural instability [49] and the rigidity of the trunk [15,18,24] have been reported to be strongly associated with disability and poor HRQoL [18] in patients with mild to moderate PD. Moreover, axial motor features of PD have been found significantly correlated with physical inactivity, decreased ability to perform ADLs and increased ADLs dependency [50]. Moreover, gait difficulty, stooped posture, rigidity and postural instability increase the risk of falls and associate consequences that greatly affect functional status and HRQoL.

In our study, the ROM of the trunk for flexion, extension and rotation movements for both sides were correlated with the PDQ-39 total score. These findings are possibly due to most functional tasks involving a trunk flexion and rotation components, such as dressing, walking, eating, transfers, bathing or turning in bed [51,52]. However, these correlations were considered poor or moderate. We did not find any associations between trunk ROM and EuroQol-5D, probably due to its generic nature. Our results could point to the fact that in early stages of the disease, flexion and extension trunk movements could be mostly affected, possibly due to increased axial rigidity. Moreover, there could be a clinical affectation for rotational movements of the trunk, and all these limitations would impact to HRQoL perceived by the patient. Nevertheless, future longitudinal studies should include fatigue scales to deepen into the relationship between trunk ROM, HRQoL and perception of fatigue.

### 4.4. Study Limitations

This study was limited by its relatively small sample size. Future studies should have a larger sample group. Our results cannot be extrapolated to other PD stages of the disease or during the “off” phase of the medication cycle. Further studies could also assess the relationship between the ROM of the trunk and axial rigidity at additional angular velocities, using other dynamic postural control tests, and examine the relationship between axial ROM impairments and HRQoL in PD patients with higher disease severity. Another limitation was the use of a non-objective evaluation of the risk of falls. Future studies should record fall incidents through longitudinal researches [53,54]. Finally, we did not investigate the relationship between trunk ROM and other relevant symptoms of PD such as freezing of gait, fatigue, posture or balance disorders.

## 5. Conclusions

Trunk ROM for flexion and extension movements, measured by a universal goniometer, were correlated with axial extensors rigidity, evaluated by a technological device at 30°/s and 45°/s and functional mobility. Moreover, trunk ROM for flexion, extension and rotations were correlated with HRQoL in patients with mild to moderate PD. However, the strengths of these associations were considered poor to moderate. These findings highlight specific opportunities for a free conventional trunk ROM measure to serve as a complementary evaluation in clinical practice.

## Figures and Tables

**Figure 1 sensors-20-02482-f001:**
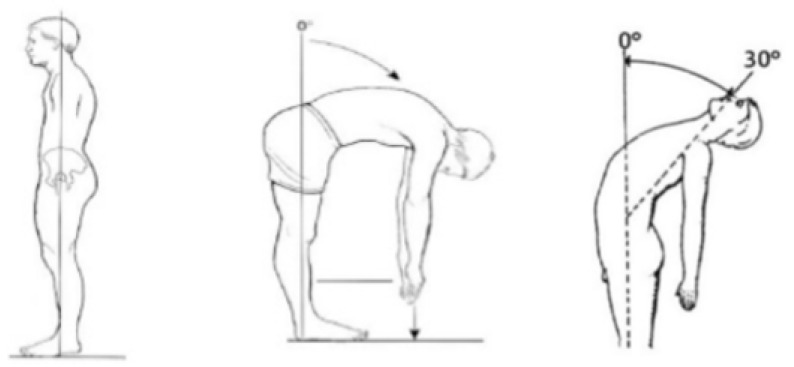
Flexion and extension range of motion (ROM) assessment.

**Figure 2 sensors-20-02482-f002:**
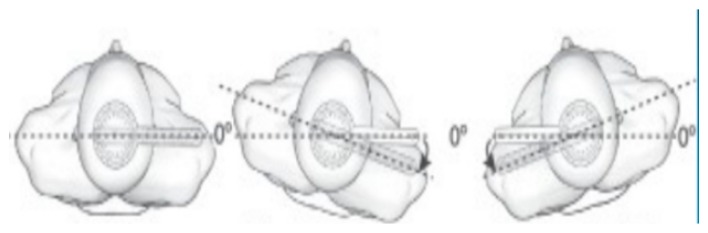
Rotation ROM assessment.

**Figure 3 sensors-20-02482-f003:**
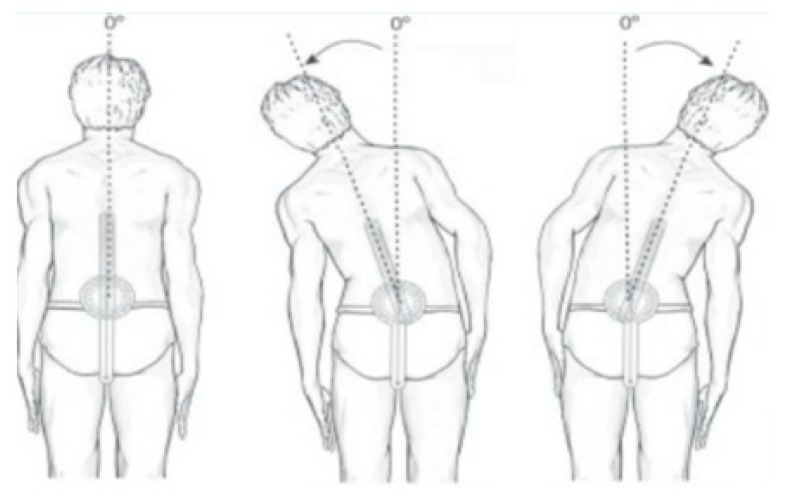
Lateral flexion ROM assessment.

**Table 1 sensors-20-02482-t001:** Clinical features of the sample with Parkinson’s disease.

Variable	Mean Score ± SD	Range
Disease duration (months)	55.4 ± 14.3	30–75
UPDRS-III *	22 ± 8	10–32
Schwab and England		80–100
80%	*n* = 26	
90%	*n* = 7	
100%	*n* = 3	
Hoehn and Yahr		I-III
IB	*n* = 8	
II	*n* = 24	
III	*n* = 4	

* UPDRS-III: Unified Parkinson Disease Rating Scale-Motor Part. Schwab and England: 80%—Usually completely independent. Takes two times longer than normal to complete chores. 90%—Completely independent. Able to do all chores but with some degree of slowness, difficulty and/or impairment. One might take two times longer than normal to complete chores. 100%—Completely independent. Able to do all chores without slowness, difficulty or impairment. Hoehn and Yahr: IB-Unilateral and axial involvement. II-Bilateral involvement. III-Bilateral involvement plus minimal difficulty walking. SD: standard deviation.

**Table 2 sensors-20-02482-t002:** Mean scores of trunk ROM, trunk rigidity, functional mobility and health related quality of life (HRQoL).

Variable	Mean Score ± SD	Range
Trunk flexion (°)	80.9 ± 13.2	70–100
Trunk extension (°)	26.4 ± 7.4	15–35
Trunk lateral flexion (right) (°)	28.9 ± 8.1	20–40
Trunk lateral flexion (left) (°)	26.6 ± 7.6	20–40
Trunk rotation (right) (°)	32.2 ± 10.7	25–45
Trunk rotation (left) (°)	30.2 ± 18.7	25–45
W/BW Extensors 30°/s	20.2 ± 13.7	10–40
W/BW Flexors 30°/s	14.9 ± 7.9	10–25
W/BW Extensors 45°/s	17.2 ± 10.1	5–35
W/BW Flexors 45°/s	11.5 ± 5.5	5–25
W/BW Extensors 60°/s	19.0 ± 11.6	5–35
W/BW Flexors 60°/s	13.7 ± 5.7	5–25
GUG		I–III
I	*n* = 14 (38.89%)	
II	*n* = 17 (47.22%)	
III	*n* = 5 (13.89%)	
PDQ-39	25.2 ± 7.0	20–40
EuroQoL-5D	70.5 ± 10.0	50–100

Values are expressed as mean ± standard deviation. W/BW: Nm°/Body Weight). W/BW Extensors: Extensor muscles trunk rigidity; W/BW Flexors: Flexor muscles trunk rigidity. GUG scores description: I—no risk of falls, II—low risk of falls, III—some risk of falls.

**Table 3 sensors-20-02482-t003:** Correlations between trunk ROM, trunk rigidity and functional mobility.

	Trunk Flexion	Trunk Extension	Trunk Rotation (Right)	Trunk Rotation (Left)	Trunk Lateral Flexion (Right)	Trunk Lateral Flexion (Left)
W/BW Extensors 30°/s	*r* = −0.687, *p* = 0.005 *	*r* = −0.534, *p* = 0.042 *	*r* = −0.032, *p* = 0.849	*r* = −0.067, *p* = 0.692	*r* = −0.156, *p* = 0.356	*r* = −0.290, *p* = 0.082
W/BW Flexors 30°/s	*r* = −0.189, *p* = 0.263	*r* = −0.017, *p* = 0.918	*r* = −0.082, *p* = 0.628	*r* = −0.067, *p* = 0.694	*r* = −0.174, *p* = 0.303	*r* = −0.015, *p* = 0.931
W/BW Extensors 45°/s	*r* = −0.510, *p* = 0.006 *	*r* = −0.410, *p* = 0.005 *	*r* = −0.030, − = 0.861	*r* = −0.025, *p* = 0.885	*r* = −0.082, *p* = 0.628	*r* = −0.180, *p* = 0.285
W/BW Flexors 45°/s	*r* = −0.103, *p* = 0.545	*r* = −0.093, *p* = 0.586	*r* = −0.267, *p* = 0.111	*r* = −0.133, *p* = 0.433	*r* = −0.062, *p* = 0.718	*r* = −0.116, *p* = 0.495
W/BW Extensors 60°/s	*r* = −0.210, *p* = 0.213	*r* = −0.091, *p* = 0.594	*r* = −0.035, *p* = 0.873	*r* = −0.043, *p* = 0.802	*r* = −0.027, *p* = 0.875	*r* = −0.192, *p* = 0.256
W/BW Flexors 60°/s	*r* = −0.027, *p* = 0.873	*r* = −0.094, *p* = 0.579	*r* = −0.168, *p* = 0.322	*r* = −0.225, *p* = 0.181	*r* = −0.153, *p* = 0.366	*r* = −0.211, *p* = 0.210
GUG	*r* = −0.444, *p* = 0.004 *	*r* = −0.564, *p* = 0.042 *	*r* = −0.651, *p* = 0.033 *	*r* = −0.677, *p* = 0.022 *	*r* = −0.189, *p* = 0.263	*r* = −0.062, *p* = 0.718

* *p* < 0.05. W/BW Extensors: Extensor muscles trunk rigidity. W/BW Flexors: Flexor muscles trunk rigidity. GUG: Get Up and Go Test.

**Table 4 sensors-20-02482-t004:** Correlations between trunk ROM and HRQoL.

Trunk Movement	PDQ-39 Total Score	EuroQoL-5D
Flexion	*r* = −0.492, *p* = 0.018 *	*r* = 0.245, *p* = 0.150
Extension	*r* = −0.549, *p* = 0.049 *	*r* = 0.265, *p* = 0.119
Rotation (right)	*r* = −0.482, *p* = 0.021 *	*r* = 0.324, *p* = 0.054
Rotation (left)	*r* = −0.447, *p* = 0.038 *	*r* = 0.249, *p* = 0.142
Lateral flexion (right)	*r* = −0.365, *p* = 0.448	*r* = 0.015, *p* = 0.930
Lateral flexion (left)	*r* = −0.476, *p* = 0.403	*r* = 0.007, *p* = 0.969

* *p* < 0.05.

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
