# Peer review of "Trunk Range of Motion Is Related to Axial Rigidity, Functional Mobility and Quality of Life in Parkinson’s Disease: An Exploratory Study"

_sensors, 2020, doi:10.3390/s20092482_

Round 1
Reviewer 1 Report
The paper presents an investigation regarding the relationship between trunk rigidity in Parkinson’s subjects and its correlation with quality of life indices and risk of fall. The paper is very interesting above all for the simplicity of the measurement methods suggested.
I think this is one of the main innovative and interesting points in this research, since a lot of investigation has been carried out and properly cited by the authors, with other more complex methods. My suggestion is to improve the visibility of such point in the introduction and, in particular, to clearly identify what the authors call technological and what is not (conventional?) according to their point of view.
Besides that I think that, in order to accept the paper for publication in Sensors, some substantial revisions are required, mainly regarding the measurement and sensor point of view, that is fundamental considering the journal the authors have considered for submission. I give hereafter just some details.
Even if a different paper presents the method, a paragraph could be devoted to recall it, for easiness of understanding. Besides that, since this Journal is devoted to sensors and measurements, in the paper I would appreciate a presentation more focused on measurements and sensors, and in results discussion some words as regards measurement repeatability (for the same subject),and measurement uncertainty (as regards this simpler method compared with the technological ones). If I have understood well authors consider only the overall standard deviation including among subjects deviations (table 2).
As regards method in 2.1 some design detail should be given: local ethical committee approval and consent are just expected information, scientific aspects should be described.
2.2 Participants: I would appreciate some considerations regarding age, and its correlation with the results ( in discussion and tables). Besides that, the quality of life indexes and even trunk rigidity could be correlated with lifestyle and with the continuous presence of a caregiver, have the authors considered such factors?
2.3 Trunk ROM assessment: is the testing sequence the same for all the subjects? If yes have the authors verified a possible bias due to the test order (flexion extension, rotation L/R and lateral L/R )? beside that have they verified a possible correlation with fatigue? Please give some details in discussion.
Table 2: Authors consider their results as scores, but actually, these results are produced by a measurement system so they should be given with their measurement units, where possible.
Table 3 and discussion: results demonstrates that trunk flexors/extensors muscles actually affect trunk flexion extension movement, but only at specific movement velocities. Please discuss in more detail these results both as regards the presence of correlation and as regards speeds at which there is no or less correlation. Are there any hypothesis about this?
The conclusion reported in discussion (line 240) is rather evident and could be expected: is this the first time these conclusions are obtained in a scientific study or is it just a confirmation? Please reconsider your discussion, rephrase it and, in the case, cite previous papers.
Reviewer 2 Report
In this study 36 patients with mild to moderate Parkinson’s disease were tested:
- Trunk ROM using a goniometer (Flex/Ext/LRot/RRot/LLatFlex/RLatFlex)
- Trunk stiffness (rigidity) using a isokinetic dynamometer (Flex/Ext) at three speeds (30/45/60 deg/s)
- Risk of falls using a Get Up and Go Test
- Health related quality of life using a PDQ-39 Questionnaire and a EuroQol-5D Questionnaire
All 6 Trunk ROM outcomes were correlated with all the other 16 outcomes, accumulating to 96 pairwise comparisons. The authors find 19 significant associations, rangeing from
r=(-)0.336 to r=(-)0.687
The authors conclude that ‘Trunk ROM deficits, as measured by a universal goniometer, are correlated with axial rigidity evaluated by technological devices, risk of falls and HRQoL in patients with mild to moderate PD.’
I agree it would be interesting to evaluate if signs for increased trunk stiffness can be obtained using a simple clinical test. However, I would recommend a complete re-analysis and rewriting of the paper for several reasons.
The authors fail to make a convincing argument why this study was performed. After the first paragraph of the introduction, I was expecting a reliability study about trunk ROM measurements. Then, in the second paragraph I become completely lost. The terms ‘rigidity’, ‘stiffness’ and ‘ROM’ appear to be used interchangeably, although they are not. There is no clear definition of these terms in the paper.
In the final paragraph the aims are introduced. The authors plan to correlate multiple outcomes (Trunk ROM, trunk stiffness, Health related quality of life and risk of falls).
The only comparison that appears to be logical to study is trunk ROM vs trunk stiffness, as the second outcome is difficult to measure in clinical practice. I would recommend to focus on these outcomes and make a clearer argument what the benefit of this information would be.
The authors appear to have been unduly influenced by the need to show results that are statistically significant at the conventional 5% level. Although that will often form a major part of any presentation, it is important to recognise that the aim of statistical methods is the elucidation of data that is affected by a multiplicity of causes. By concentrating solely on results that are statistically significant the authors are failing to look at the bulk of the data.
The first question I would have asked would be what is the relationship between left and right trunk rotation, and what is the relationship between left and right lateroflexion, and what is the relationship between the trunk stiffness estimates at different velocities. If they are very highly correlated I would have reported one outcome for each (e.g., total rotation range of motion, or average stiffness estimate over all velocities).
My next stage would have been to perform multiple regressions of the remaining ROM variables on each remaining stiffness outcome. In the first instance I would be reporting the significance of that regression, rather than the significance of the individual independent variables. That approach avoids the problem of multiple testing in relation to the independent variables.
Multiple testing with respect to the dependent variables cannot be avoided but I would be very cautious in the conclusions I drew. I would emphasise that this is a hypothesis generating study rather than a hypothesis testing study. In relation to this aspect of multiple testing it is not valid to choose at this stage to omit some of the variables that have been analysed, so I would reinstate all of the your original variables.
Reviewer 3 Report
The authors performed a cross-sectional study on 36 PD patients with a mild to moderate disease stage (Hoeh and Yahr score: Ib-III) to investigate for the first time the correlation between many features of the trunk ROM (as per a clinical goniometer evaluation) with axial rigidity (as per an isokinetic dynamometer), risk of falls (as per the clinical Get up & Go test), an quality of life (by two validated questionnaires). The research question behind the study and the topic is of relevance for the field of movement disorders. The method applied sounds quite rigorous, and the manuscript is clear and well-written. However, this study should be categorized as explorative, and this aspect should appear in the title, the abstract and the methods section for the following reasons: the authors tested many variables in a small sample size using a very simple statistic, allowing to uncover some correlations, without any clues on cause-effect or predictivity. Moreover, testing for many variables in this way, there are some findings not having a clear explanation, e.g., why trunk rotation ROM on the right should be more associated with QoL than the one on the left?
A second relevant concern is about the very clinical nature of this study (which I appreciate), with the use of a technological device only as one of the outcome measures. Might this article be more suitable for a clinical neurological or rehabilitative journal?
Below my further comments and concerns:
- I would suggest a ref 1 more pertinent to the context.
- Page 2, line 78. ‘Mini-Mental State Examination scores ≤ 23 points’. I suppose this is an exclusion criteria instead of an inclusion criteria. If so, I suggest to ‘justify’ the choice stating that patients with dementia, established by a MMSE score ≤ 23 points’, were excluded.
- Page 2, line 79. Please, explain why depression was an exclusion criterion and how do you assess this comorbidity.
- Please, add information on the possible presence static postural abnormalities (such as camptocormia, pisa syndrome, antecollis, or milder forms) of included patients.
- A picture clearly showing examples of the use of the universal goniometer in the trunk ROM would be useful.
- It would be useful to add ranges along with mean and SD in Table 1 and 2 where applicable
- In the discussion section, the authors should try and find an explanation to the different associations found between right and left side rotations or flexions.
Round 2
Reviewer 2 Report
The authors clearly improved the manuscript within in a brief time window. Although I am very sorry for the authors with respect to the current COVID-19 situation, which prevents them from rerunning statistical analyses, this does not change my point of view in this regard.
The authors should present a concise and logical plan for their statistical analyses, rather than correlating (almost) all outcomes, resulting in 96 pairwise comparisons.
For example, the authors report a significant correlation between right trunk rotation ROM and PDQ social support. What makes the relation between these outcomes relevant to know? And why specifically right rotation? In any cohort of patients one would expect to find significant correlations between the different factors of the international classification of functioning; people who experience pain or reduced joint mobility, will be able to perform less activities, which affects their environment, personality and QoL as well. This relation between factors will probably be even stronger in systemic problems such as Parkinsons. This does not mean that these outcomes are causally related (i.e., right trunk ROM does not affect social support or the other way around).
Because the statistical analyses lack so much focus, it is highly likely that some significant correlations are significant by chance rather than as a result of a true association. Moreover, most significant correlation coefficients have a value of around 0.4, which corresponds to a common variance of 16%. That is a rather weak association. This does not resonate through the discussion, conclusion and abstract.
Finally, the authors should be far more preservative with their use of the words 'risk of falls' as they did not record fall incidents, but a get up and go test. The authors provide no convincing evidence that this has a strong relationship with fall risk.
Reviewer 3 Report
The authors have addressed all my comments and concerns. I think the quality of the paper has improved and I do not have further comments. Minor text editing would be necessary to fix some typos.
Author Response
Thank you very much for your comments, we really appreciate them. We think that the changes have substantially improved the paper.
Round 3
Reviewer 2 Report
-